# Freshwater as a Sustainable Resource and Generator of Secondary Resources in the 21st Century: Stressors, Threats, Risks, Management and Protection Strategies, and Conservation Approaches

**DOI:** 10.3390/ijerph192416570

**Published:** 2022-12-09

**Authors:** Doru Bănăduc, Vladica Simić, Kevin Cianfaglione, Sophia Barinova, Sergey Afanasyev, Ahmet Öktener, Grant McCall, Snežana Simić, Angela Curtean-Bănăduc

**Affiliations:** 1Applied Ecology Research Center, Faculty of Sciences, Lucian Blaga University of Sibiu, I. Raţiu Street 5–7, 9, 550012 Sibiu, Romania; 2Department of Biology and Ecology, Faculty of Science, University of Kragujevac, R. Domanovića 12, 34000 Kragujevac, Serbia; vladica.simic@pmf.kg.ac.rs (V.S.); snezana.simic@pmf.kg.ac.rs (S.S.); 3FGES, Université Catholique de Lille, F-59000 Lille, France; kevin.cianfaglione@gmail.com; 4Institute of Evolution, University of Haifa, Mount Carmel, 199 Abba Khoushi Avenue, Haifa 3498838, Israel; sophia@evo.haifa.ac.il; 5Institute of Hydrobiology National Academy of Sciences of Ukraine, Prospect Geroiv Stalingradu 12, 04210 Kyiv, Ukraine; safanasyev@ukr.net; 6Ministry of Food, Agriculture and Livestock, Food Control Laboratory Directorate, Denizli 20010, Turkey; ahmetoktener@yahoo.com; 7Center for Human-Environmental Research (CHER), New Orleans, LA 70118, USA; mccall@cherscience.org

**Keywords:** freshwater, stressors, threats, risks, management, protection, conservation

## Abstract

This paper is a synthetic overview of some of the threats, risks, and integrated water management elements in freshwater ecosystems. The paper provides some discussion of human needs and water conservation issues related to freshwater systems: (1) introduction and background; (2) water basics and natural cycles; (3) freshwater roles in human cultures and civilizations; (4) water as a biosphere cornerstone; (5) climate as a hydrospheric ‘game changer’ from the perspective of freshwater; (6) human-induced stressors’ effects on freshwater ecosystem changes (pollution, habitat fragmentation, etc.); (7) freshwater ecosystems’ biological resources in the context of unsustainable exploitation/overexploitation; (8) invasive species, parasites, and diseases in freshwater systems; (9) freshwater ecosystems’ vegetation; (10) the relationship between human warfare and water. All of these issues and more create an extremely complex matrix of stressors that plays a driving role in changing freshwater ecosystems both qualitatively and quantitatively, as well as their capacity to offer sustainable products and services to human societies. Only internationally integrated policies, strategies, assessment, monitoring, management, protection, and conservation initiatives can diminish and hopefully stop the long-term deterioration of Earth’s freshwater resources and their associated secondary resources.

## 1. Introduction and Background

Due to its unique and profoundly necessary characteristics, water is the key element and component in the origin, continuity, and evolution of life on Earth. Water is everywhere, even in living organisms, and for that, it is a key to the study of ecology and biodiversity, to preserving the environment, and to understanding the universe [1,2].

Water is an extremely important substance for all life (including humans) on our planet. Water exists in large amounts, as there are around 4.16818183 km^3^ of water on Earth in three its three physical states: solid, liquid, and gas. It differs in its abundance across the following contexts: seas and oceans (97.24%), glaciers and icecaps (2.14%), groundwater (0.61%), freshwater lakes (0.009%), inland seas (0.008%), soil (0.005%), atmosphere (0.001%), and lotic systems (0.0001%) [3].

Water may seem bountiful, but only under 1% of the Earth’s water can be used to meet human demands [4]. Human population growth is a dominant contributor to water scarcity and decreasing water quality, and it is an obvious fact that the world’s human populations are growing [5]. Human requirements for water and demand of limited water resources, among other things, will exacerbate this set of problems [6]. Restrictions on water availability are among the most important reasons for conjecturing about a future limit to the human activities and growth of the world population.

Climate modification, which is very much tied to human population increase, will induce huge pressure on water resources. The warming of the global climate system is undeniable because of an uninterrupted warming trend observed since the mid-20th century, which can be mainly ascribed to human-induced anthropogenic influences [7]. Therefore, there is a strong consensus that freshwater systems are closely interrelated with climate and weather variation. In this way, global, regional, and local climate change may have direct influential effects such as water temperature increases, decreased dissolved oxygen content in water, increased toxicity from pollution, etc. [8].

Demand for freshwater is rising everywhere in concert with factors such as population growth, climate fluctuation, land use change, etc., which render future water availability uncertain [9]. This also highlights the need for state-of-the-art research on water resource threats, risks, trends, and management.

Some view water as a hostile habitat stemming from our terrestrial way of life, and this is why our rather basic knowledge of these habitats has led to severe difficulties for humankind. This includes wetland vegetation destruction, reclamation, water overuse, and pollution. Biological resources and ecological services are also threatened [10], and this all highlights an urgent necessity for a much greater awareness of freshwater bodies and the natural processes that are happening to them and that sustain our human ways of life based on it. A proper local, regional, and global inventory of the uses of freshwater by humans can raise awareness of issues, including the following: drinking water, waste disposal, shipping and transport, provisioning of human food, irrigation, power generation, recreation, etc. There is also an expansive list of water-related problems that can offer us a rather complete image of looming disasters: lack of safe drinking water in developing countries; the related lack of access to sanitation facilities; the impacts of freshwater crises due to climate change on agricultural production; negative impacts of decreasing water flows on main rivers in terms of transcontinental transport; harm to aquatic and semiaquatic ecosystems (biomass production, diminishing genetic variability, hampering the ecological status of wildlife, diminishing industrial water accessibility, etc.).

Starting in the 20th century, UNESCO has made an obvious commitment to work with all potential partners to alleviate the water-related problems that the world is facing. The Millennium Development Goals also target sustainable access to safe drinking water and sanitation. The Johannesburg Summit challenged the world to provide safe drinking water and sanitation by 2015, and to do so in terms of ecosystemic integration. Where are we now? Can we manage freshwater and other associated resources in sustainable way if we are forced look honestly at the present context of the vulnerability of this resource? Unfortunately, it is obvious that, at least in the near future, we are not even close to meeting this challenge.

This paper provides a synthetic overview on some of the threats, risks, and integrated water management issues in freshwater ecosystems and its implications for human needs and freshwater conservation. Below we review the following: (1) introduction and background; (2) water basics and natural cycles; (3) freshwater roles in human cultures and civilizations; (4) water as a biosphere cornerstone; (5) climate as a hydrospheric ‘game changer’ from the perspective of freshwater; (6) human-induced stressors’ effects on freshwater ecosystem changes (pollution, habitat fragmentation, etc.); (7) freshwater ecosystems’ biological resources in the context of unsustainable exploitation/overexploitation; (8) invasive species, parasites, and diseases in freshwater systems; (9) freshwater ecosystems’ vegetation; (10) the relationship between human warfare and water.

In spite of the fact that global water resource management is extremely complex, owing to diverse geophysical, climatic, socioeconomic, and political realities, the core of this approach should be the belief that only the conservation of complex ecosystem structures and functions can solve the present global water crisis. The importance of this review article, then, is to highlight a series of principal stressors with synergic effects, which must be considered in terms of the assessment, monitoring, and management of complex freshwater ecosystems.

## 2. Water Basics and Natural Cycle

The word water comes from the Old English word waeter, the Old Saxon water, and Proto-Germanic watr [11]. Water is the only commonly occurring substance on Earth to exist as a solid, liquid, and gas within the range of normal terrestrial conditions [12], constituted as rain, snow, ice, rivers, lakes, and oceans. In its pure state, water is a transparent, odorless, tasteless and nearly colorless liquid substance with a hint of blue. As a compound of hydrogen and oxygen, H_2_O, water freezes at 0 °C/32 °F [13].

There are few main hypotheses for the origins of Earth’s water, which are based on our planet’s retention of water in some form throughout the process of its accretion through major impact events [14]. Another scenario involves the formation of the Moon [15], possibly including an interplanetary collision that led to the Moon’s formation [16].

Water, a polar inorganic compound, includes one oxygen and two hydrogen atoms. It is the most widespread molecular compound of our planet [17] and a universal solvent due to its capacity to dissolve more substances than any other liquid [18]. Its properties as a solvent allow water to play a crucial role in the development of life [12].

The water cycle is widely taught as an uncomplicated cycle of evaporation, condensation, and precipitation. The reality, however, is much more complex. Both the natural and human-induced changes to freshwater quantity and quality have become a major issue of concern [19] given that the natural paths and influences on water through Earth’s ecosystems are highly complex at local, regional, and global scales [20]. For the first time in history, a global freshwater crisis has occurred, which is partly attributable to economic globalization and human-induced perturbation of natural resource commodities, including the water cycle [21].

The natural influence of the water cycle on ecosystems’ and biomes’ ecological status, resilience, and productivity, are harmed by a great number of human-induced causes (human population growth, economic development, etc.) and impacts (climate modifications, physicochemical pollution, habitat fragmentation, biological resource mismanagement, invasive species, parasites, wars, etc.). This has led to insecurity across many human socioeconomic domains regarding water for drinking, industrial applications, irrigations, hydropower, waste disposal, recreation, etc. These stresses have been accentuated by climate variations that affect the hydrologic cycle [10,11,12,13,14,15,16,17,18,19,20].

In many parts of the world, the ecologic assessment of water bodies is purely aspirational. Furthermore, even in those parts of the world where we believe that water bodies have been properly assessed from an ecological standpoint, we often discover that such studies were too narrow in their scope and scale [22,23].

Water resources are threatened in terms of both quantitative and qualitative dimension of risk. The very uneven distribution of global freshwater quality is affected by a huge number of physical, chemical, biological, and ecological elements and factors. Quantitative aspects are influenced at the global scale by various aspects of the water cycle and locally/regional by human-induced changes such as human overexploitation [22,24,25,26].

The unequal distribution of freshwater is also a main reason for water overexploitation and abuse, given that about 60% of water flowing in all rivers worldwide is shared by two or more countries [27].

All of the above are not only potential risks but are also, as part of an escalating water crisis increasing worldwide, aspects of important long-term consequences felt either directly or indirectly by human societies [24].

The scarcity of freshwater is an increasingly critical problem in many parts of the world. Water quality and quantity can mutually reinforce or affect one another, which necessitate a joint management strategy [28,29]. Complex integrated local, regional, and global management strategies are urgently needed, as well as specific measures that can be created and implemented. Action is needed before ecological, economic, social, and political disputes appear and grow, in order to avoid international tensions and even potential conflicts ravaging the planet [30,31].

## 3. Freshwater Roles in Human Cultures and Civilizations

We can declare that the human civilization as a whole was built and maintained based on water use. The world’s major cultures and civilizations arose and developed in the proximity of freshwater bodies, which draws attention to their significance. Over 10,000 years ago, when humans entered the agricultural revolution and adopted an agrarian way of existence, humankind began living in permanent settlements, all of which were very much dependent on water. This created a brand new type of relation between humans and water. Pathogens transmitted by contaminated water from urban and agricultural areas became a very important health risk for sedentary human communities. In these circumstances, guaranteeing a sufficient amount of clean water for peoples’ needs become a precondition for the appearance and expansion of urbanization, state formation, and other developments in human social organization that came afterwards, all of which are related to huge changes in human demographics [32,33].

The relationship between human civilizations and water-associated risks meant that people began to learn to understand and survive with them. This process included the modification of their perceptions, beliefs, attitudes, and behaviors that constituted their broader ways of life [34]. In one way or another, water-related stressors and/or catastrophes deeply influenced some major driving cultural and religious beliefs: ancient Sumerian legends (3000 BC) recount the deeds of the deity Ea, who punished humanity for its sins by inflicting the Earth with a six-day storm. This Sumerian myth parallels the biblical account of Noah and the deluge, although some details differ. In Egypt, Hapi was a principal god of the annual flooding of the Nile and played a major role in both religious beliefs and ceremonies and the theological justification of pharaonic rule [35]. Other such examples are too numerous to list here.

Beyond its role in incipient agricultural surplus production, water was also used for other special uses that changed human history. Such an example is the use of water as a weapon. For example, the Sumerians dammed the Tigris River between 1720–1684 BC, the Spartans poisoned the cistern of Piraeus/Athens in 430 BC, and Caesar and the Romans constructed ditches at the siege of Alesia/Gaul in 52 BC. In 1904, the German colonial troops poisoned desert wells in 1904 to defeat Herero rebels in Africa. In using water as a defensive weapon, Moses and the Jews parted the Red Sea waters and closed it behind them in 1200 BC. Nebuchadnezzar and the Babylonians used the Euphrates River as a defense between 605–562 BC. Saladin and Arabs cut off the Crusaders’ wells’ water in 1187. In the Russian–Ukrainian war in 2014–2022, water pipelines have been intentionally damaged, leaving millions of people without water in Ukraine [35].

Water has also been a trigger for numerous violent interactions and conflicts through history. Pontius Pilate started a conflict between the Romans and Jews due to a stream diversion in Jerusalem in 30 AD. In the U.S. state of Texas, cattlemen fought landowners for access to water in 1870. In 1898, France and Britain battled over the Nile. In 1958, Egypt and Sudan clashed over the Nile. In 1963–1964, Ethiopian and Somali nomads fought for desert water. In 1970–1991, militias from competing Chinese localities fought over water extraction. In the 1980s–1990s, there were many military encounters between Cameroon and Niger over the recession of Lake Chad. In 1990–1992, violence over water competition killed hundreds in Uzbekistan and Turkmenistan. In 1992, Hungary and Czechoslovakia had a dispute over the Danube. In 1995, Ecuador and Peru fought over the Cenepa River. In 2001, Macedonia and Albania fought over the control of water supply reservoirs. In 2019, in Mali, 50,000 people fled their homes as conflict over water escalated [35].

Since the foundation of the earliest human villages, towns, and cities, there has been and still is an imperative bond with freshwater bodies. Jericho is one of the earliest recognized permanent settlements, dating to around 9000 BCE, roughly the beginning of the Holocene. Jericho was strategically placed near a water source, grew into a major early city, and persisted for thousands of years. In Mesopotamia and Egypt, there is evidence of wells and stone rainwater channels beginning around 3000 BC. From the beginning of the Bronze Age, in the city of Mohenjo-Daro, located in modern Pakistan, hundreds of ancient wells, water pipes, and toilets have been found. Similarly, the first European water control systems, including baths, toilets, and drainage, were constructed in Bronze Age Crete in the second millennium BC. The Roman Empire employed wonderful water systems centuries before far less effective water facilities of medieval cities [36].

After the Dark Ages, ease of access to water, quality, hygiene, and dealing with epidemics improved. Essential modifications materialized: science, know-how, and sanitation were institutionalized as modern universities appeared during the 13th century. Agriculture began to industrialize in the 18th century. With industrialization and urbanization, there was extraordinary progress in terms of transport and commerce, and a boom in terms of the world human population. This sparked increasing globalization and an unrealistic desire for permanent limitless development. It put a terrific stress on the water as a resource [36] and of a source of economic production.

Throughout history, there have been various ingenious solutions with which to guarantee an ample amount of clean water for human society; however, such technology slowly began to approach a kind of dead end. In terms of its ecological consequences, our knowledge, perspectives, approaches, etc., must be improved. How do we do this? Clearly, it involves further assessment, monitoring, study, management, and governance. These concerns have to do with the role of water in terms of climate, habitats, species, biocoenosis, ecosystems, biomes, and ultimately the biosphere.

As long as human population growth is a dominant contributor to freshwater scarcity and decreasing quality—and it is a fact that the world’s population is growing steadily—the requirements for freshwater and demand on limited freshwater resources with be exacerbated, and the risks for more and greater water conflicts will expand.

Across these local and regional issues, the objective limits of the accessibility of freshwater and its utilization are among the central reasons for hypothesizing a limit to human activities and the long-term increase.

## 4. Water a Biosphere Cornerstone

The biosphere is the rather thin life-supporting layer of Earth’s surface, forming extremely integrated complexes of ecosystems, from a few kilometers into the atmosphere to the deep-sea hydrothermal vents, as well as soils and geological substrata. Water plays vital roles in the operation of the biosphere. Freshwater ecosystems are essential, providing a varied and crucial collection of products and services upon which human society depends [37].

Since the quality of world fresh water is influenced by a huge number of physical, chemical, biological, and environmental elements and factors, including climatic features [38] as well as various human activities, it is necessary to determine what the boundaries of the ecosystem are and how the limits of fluctuation in the values of its elements correspond.

Aquatic ecosystems can be formed in the range of environmental variables only because the water on Earth appears with different characteristics in terms of chemical dilution. In this way, each dissolved element in a life-supported ecosystem has its own range of concentrations [39] and organisms surviving in it, which is a well-documented feature of the Sládeček model [40]. Primary producers such as algae and cyanobacteria are placed on the base of the trophic pyramid in aquatic ecosystems [39]. Thus, the existence of a trophic pyramid and its structural elements are determined at the phase of the separation of inorganic compounds (nutrients) and biological organisms (photosynthesis). However, since the aquatic ecosystem exists only within certain limits in terms of the inorganic environment, it is important to identify the state (species specificity, abundance, and biomass) of the first trophic level in order to determine the structure of the ecosystem.

Threats and risks to aquatic ecosystems are thus related to its structure, which is based on the activity of photosynthetic organisms. It is they who primarily respond to impacts and changes in the environment, which results in changing their abundance and biomass. That is, it changes the structure of the ecosystem. In order to assess the value of the acceptable impact on an aquatic ecosystem, it is necessary to understand exactly how and in relation to what stressors the ecosystem reacts; that is, to determine its structure and its buffering capacity.

Stressors that catastrophically affect the aquatic ecosystem are primarily floods and the introduction of harmful materials to water, whether through a natural disaster or anthropogenic activity. These impacts are regulated in developed countries, where it is possible to control the influencing factors, especially as a result of anthropogenic interference [41].

Because surface water is divided into two types—standing water (lakes, marshes, and swamps) and flowing water (rivers and streams)—the impact on each type of stressor is also different. The effects of pollutants such as trace metals, herbicides, or hydrocarbons on freshwater habitats vary according to the rate of water flow and the habitat’s specific characteristics. Therefore, flood rates must be regulated as a key factor related to impacts [41].

The impact of changes in the concentrations of elements dissolved in natural waters is regulated according to two principles: (1) regulation of the concentrations of wastewater into natural water bodies and (2) assessment of the existing concentrations in the water body. The first principle only defines the threshold beyond which is punishable by fees and fines. The second principle is to determine not only the threshold but also the gradation of certain substances according to 3–5 water quality classes.

Within the PFD, there are intercalibrated concentrations of dissolved substances that are accepted as quality standards in each specific country [42]. However, judging by the threshold values [43], all waters should be no higher than Class 2 in terms of quality, which is never or very rarely observed in nature, even in undisturbed contexts, since there is a process of species succession and ecosystem evolution.

One of the most significant problems, but also the trickiest to interpret its parameters, is organic pollution. The allotment of water parameters into five classes [44] is consistent with the categorization of indicators of organic pollution in the model of V. Sladechek [40]. This provides the basis for the use of indicator organisms to find out the water quality class. This system gives a complete assessment of the state of the aquatic ecosystems, regardless of its type—river or lake. It is broadly used, as it allows for exposure of the rank concentrations of the main environmental variables, as well as the chemically indeterminate indicators of the type of nutrition in an ecosystem and its trophic state [45]. In view of the prospects of this method, which makes it possible to determine the ranking of environmental parameters by the composition and abundance of species in an ecosystem, it is recommended for use in monitoring systems in both European countries and the United States, along with the hydrochemical and hydrophysical properties of waters [46], including bioindication and pollution indices within the framework of the WFD [47].

The interactions between algal biodiversity and the environment are induced by the adaptation level of the species and the community they form. Bioindication relies on the principle of congruence between community composition and complexity of environmental elements [45]. On the other hand, it is still difficult to describe the role of particular environmental variables, and to forecast the community’s response to environmental changes. Governments, scientists, managers, and the general public must pay attention in evaluating the health of ecosystems [48]. Consequently, the difficulty of the evaluation of an ecosystem state and the prediction of its changes is still up to now an issue of understanding how we could evaluate the threats and risk factors and to manage them [49].

The rate of response of biotic systems to the impact of a stress factor depends on the level of the biota’s organization [49], and for autotrophs, the response is noticeable at the scale of days to years. However, it is the highest levels of diversity that are distinguished by sufficient buffering capacity in terms of ecological impacts, and are most relevant to the broader ecological situation in a water body.

The monitoring of biological elements is a main aspect of the evaluation of the chemical and ecological status within the Water Framework Directive. These broadly accessible biological methods for assessing the water quality-based definition of biological conditions have been described previously [50]. The progress of forecasting models as a tool for the evaluation of the state of aquatic ecosystems is great [51]. Ecological modeling has a long history from hierarchical [52] to structurally dynamic (SDMs) models [53]. As a result, the ‘ODD’ (overview, design concepts, and details) protocol [54] was produced with the goal of standardizing the published descriptions of individual-based and agent-based models. Over the last few years, models have been designed, including GIS, neural networks [55], self-organizing maps [56], and spatial visualization maps [57,58].

Such indexes as the saprobity Index, the Shannon Index, and species richness can be described as the main succession trends with the trophic base increasing [39]. The levels of saprobity indices are correlated with the environmental data from the Sládeček model [40] (having to do with water quality classification variable ranges [39] and the major successional stages in algal communities). Thus, they can be described as community parameters that reveal a natural intact community and a self-purified ecosystem range of variables, as well as vital parameters up to variables in which the ecosystem can collapse. The key successional stages in the model coincided with the self-purification zones in the Sládeček model [40]. Such models work with data restricted to a particular water body, and consequently, they can forecast the development of only this specific system and just for a short period of time. Such a model can be used as a predictive tool for a large period for any aquatic ecosystem, taking into account all biological and environmental data in the frames of which the aquatic ecosystem can be present.

Estimates based on the autoecological data of species from different trophic levels are consistent, but communities of organisms of a higher trophic level also show higher trophic estimates. For example, simultaneously collected phyto- and zooplankton give similar estimates in terms of the state of the aquatic ecosystem; however, zooplankton saprobity indices are slightly higher [59].

Thus, the use of indicator organisms for the hydrochemical and hydrophysical properties of water makes it possible both to refine the assessment of the state of the aquatic ecosystem up to five classes and nine ranks, and to expand the estimated parameters to organic pollution, types of nutrition, trophic status, risk of negative (toxic) impact, stability, and other predictive properties. Currently, the number of indicator species has expanded to more than 8000 algae and cyanoprokaryotes [60], 1700 invertebrates [61], and about 300 aquatic plants and mosses [62]. This represents a wide choice of organisms for assessing the parameters of the ecosystem of a reservoir, wherever it is located and whatever its status.

Based on the above premises, it seems possible to identify the main management elements that, with an appropriate management scheme, could improve the quality of water resources. First of all, hydrological stability should be observed and measurements should be taken to prevent floods and irreversible anthropogenic impacts, which are already regulated by documents within the framework of the FWD. Further definition of the levels of the trophic pyramid and water indicators should be determined for a particular reservoir, its type, climatic zone, country, and standards adopted in it for monitoring and in assessing water quality. In the framework of the indicators of ecosystems proposed above, in any case, the state and diversity of the biotic component of the reservoir, such as species diversity, structural indices, and pollution indices based on indicator species, should be observed. In this regard, there is a need to create regional databases of indicator organisms for the surface waters of each country [63]. Here, the problem of assessing the transboundary impact and the impact of climatic parameters of each specific country is revealed [38].

While in terrestrial ecosystems it is possible to protect not only a part of the landscape, but also a specific tree, the protection of aquatic ecosystems is associated with an understanding of what and where we can save. Most often, the question arises with the protection of specific species; red books are developed in accordance with the threat criteria.

The conservation status and rarity of aquatic inhabitants can be assessed according to the International Union for Conservation of Nature (IUCN) criteria [64]. For macrophytes, including charophytes and aquatic animals, this approach is justified only if the habitat of the species population supports its non-extinction. For small cell populations of aquatic organisms, which in any water body is responsible for the disposal of incoming pollutants, such an approach is impossible or unproductive. In any case, to protect against a population, it is necessary to protect not only individuals, but also their habitat, i.e., the aquatic ecosystem. This brings us to the Sladeček model, which defines the scope of classes of aquatic ecosystem parameters and their corresponding utilization in the classification of water quality. In any case, a framework of ecological values for the protected species should be defined and then a mechanism for regulating water use should be put in place to ensure that water quality does not deteriorate. Control should determine the change in the water parameters necessary for the survival of a particular species, but also in the successional stage of the ecosystem it needs. Consequently, the protection of a species takes place in three stages: (1) identifying the optimal environmental parameters for the protected population, (2) monitoring these indicators, and (3) monitoring the conservation and nondeterioration of the parameters of the ecosystem of this species, because otherwise, the species falls into a higher threat category. In this regard, the actions of both parties should be coordinated in the case of a transboundary watercourse that includes the range of a protected species.

## 5. Climate as a Hydrosphere Game Changer, a Freshwater Perspective

The hydrosphere is the irregular layer of water at or near our planet’s surface. It includes all liquid and frozen water, geological and pedological groundwater, and vapors in the atmosphere [65]. Climate refers to the conditions of the atmosphere at a specific place over a long time period. It is the long-term summation of the atmospheric elements (solar radiation, temperature, humidity, precipitation, pressure of atmosphere, air mass movement, etc.) that, over short periods, form weather. [66].

Climate modification is one of the foremost recognized crises, including all natural and anthropogenic conditions and circumstances [67,68,69,70]. The prognosis realized based on global climatic modeling simulations brings to light the fact that the main critical elements can be both natural (fluctuations in solar radiation, volcanic eruptions, and aerosol aggregations) and human-induced (variation in the composition of the atmosphere due to anthropogenic activities). Only the synergic effect of these two categories of elements can contribute to the modifications revealed in Earth’s global temperature in the 20th and 21st centuries [71].

It is generally agreed that inland water states are intimately related with weather variations, so climate change may have a powerful influence on freshwater habitats and biocoenosis [72]. The IPCC Climate Report, entitled “Code Red for Humanity”, gives priority to indisputable evidence about the fact that warming has accelerated in the last few decades. The Earth’s warming is influencing all the areas of our planet and further warming is forecasted for the next many decades. Numerous factors related to human activities and feedback systems will no doubt aggravate this set of problems [73]. Another result of climate modification is hydrologic cycles, with growing force and repetition of excessive events such as droughts as a significant example. This trend could influence freshwater habitats and biocoenosis, influencing phenology, life cycles, and distribution, and in some cases, the extinction of hypersensitive plant and animal species [74]. Increasing climate modification is predicted to have consequences for the biodiversity of extremely large areas, with impacts on the dispersion and existence of numerous plants and animal species, and the disappearance of various ecosystem types [75]. There is rich evidence that planetary heating is putting at risk the freshwater biodiversity of the Earth [76].

In this planetary heating sequence of events, freshwater habitats and biocoenosis are extremely exposed and their species could experience huge impacts. Studies have revealed that freshwater biodiversity has decreased faster than marine and terrestrial diversity [77]. The Earth’s climatic warming is beyond doubt due to the almost continuous heating tendency since the 20th century, which may be related to the impact effects of human activity [78,79].

Drought is a consequence-dependent event [80], and due to significant anthropogenic impacts [81], induces a high hydroclimatic risk for natural and anthropogenic system elements [82]. Added to this, heat waves are predicted to increase in frequency in the future [83].

Climate modification-related issues are some of the most intriguing problems to be addressed in our time. In the present climate change context, temperature increases are everywhere [84], even in surprising areas of the planet [85,86,87,88], and drought is a main driver for freshwater ecosystems’ ecological state [89], aquatic biodiversity [90], and their economic uses [91,92], even in what are considered ‘secure’ zones.

## 6. Human-Induced Stressors’ Effects on Freshwater Ecosystem Changes

There are countless stressors and associations that act as drivers for freshwater ecosystem changes. Here, we discuss a few of them that are considered to be very important in this context.

### 6.1. Pollution

Pollution can be defined as the addition of any substance (solid, liquid, or gas) or any form of energy (heat, sound, radioactivity) to the environment at a rate faster than it can be dispersed, diluted, decomposed, recycled, or stored in some harmless form [93]. Persistent organic pollutants (POPs), which pose a potential threat to ecosystems, as well as human safety and health [94], are organic chemical compounds with toxic characteristics. They are resistant to decomposition and can bioaccumulate in living organisms and be moved and carried by air, water, and migratory organisms for very long distances [95]. In general, according to the Stockholm Convention, the POP family can be divided into intentionally and unintentionally produced POPs, legacy POPs, and newly-listed POPs (emerging POPs).

POPs can have two categories of origin: dibenzofurans and dioxins, which are chemical compounds released by volcanic manifestations and fires [96,97], or as the result of human activities, e.g., different chemical substances used in agriculture, such as pesticides (hexachlorobenzene, chlordane, heptachlor, DDT, aldrin, dieldrin, endrin, mirex, toxaphene, etc.). The latter includes a wide variety of industrial chemical substances (perfluorinated compounds, brominated compounds, polychlorobiphenyls, hexachlorobenzene, etc.) and by-products of industrial activities or burning (dioxins and furans) [98,99,100,101].

An organic substance Is considered a POP if it meets the following criteria: it is persistent, with a half-life in water of over 60 days, over 2 days in air, and in sediment over 180 days [102]; it has a high transport capacity at long distances by water, air, and migratory organisms [103]; it has bioaccumulation and bioamplification characteristics [104,105,106]; and it is noxious, with negative impacts for both the human health and the environment [94,107].

As lipophilic substances with high stability through time, POPs bioaccumulate, bioamplificate, and concentrate throughout all freshwater trophic levels [108,109,110]. Furthermore, they become toxic at some specific concentrations [98,104,111,112,113] for humans as well as ecosystems [114]. Due to of all these features, POPs are considered among the high-risk stressor pollutants to natural, seminatural, and anthropogenic environments, as well as human health.

### 6.2. Habitat Fragmentation, Contraction, Destruction, and Loss

Habitat fragmentation is the process through which a large area of habitat is changed into smaller patches of a total area cut off from each other by a matrix of habitats differing from the original [115]. Habitat contraction/destruction/loss is the process by which a natural habitat becomes incapable of sustaining its native species and cannot offer is characteristic natural products and services to human society [116].

#### 6.2.1. Habitat Fragmentation and Loss in Lotic Systems

Lotic systems are flowing waters that drain the landscape and include biotic interactions amongst organisms as well as the abiotic physical and chemical interactions of its many parts [117].

Lotic systems offer extremely important natural services and products such as hydrological regulation (i.e., groundwater recharge and discharge, storage and retention of water for domestic, industrial, and agricultural use), pollution control and detoxification (i.e., retention, recovery, and removal of excess nutrients and pollutants), erosion (retention of soils and sediments), water purification, groundwater recharge and discharge, climate change mitigation (i.e., regulation of greenhouse gases, temperature, precipitations, chemical composition of the atmosphere), natural hazard mitigation (flood control, storm protection), abiotic (i.e., firewood, peat, minerals) and biotic (i.e., fish, mollusks, medicine, ornamental species) elements, cultural elements (spiritual, inspirational, recreational, aesthetic, educational values), and supporting soil formation, nutrient cycling, pollination, etc. [118].

Lotic systems’ fragmentation, contraction, alteration, and loss are key stressors impacting aquatic environments in terms of abiotic and biotic elements, and can be produced by the following: human population growth; human consumption growth, infrastructure development (dams, dikes, levees, diversions, interbasin transfers, hydropower systems, poaching, etc.); land conversion; overharvesting and overexploitation; introduction of exotic species; release of pollutants into the water, land, and air; climate change, etc. [86,119,120,121,122,123,124,125,126,127,128,129,130,131,132,133,134]. These are still a largely unquantified threat [135]. These all amount to a vast and intense combination of environmental pressures that have emerged among poor ecological conditions and have contributed to the extinction of many endemic sensitive species.

There are well-known periods in Earth’s history in terms of hydroclimate that are both flood-rich and flood-poor [136]. The significance of fluctuations in freshwater natural water flows to the long-term sustainability and productivity of aquatic, semiaquatic, riverine and riparian areas is that these fluctuations are characterized by temporal and spatial heterogeneity in the scale, occurrence, period, timing, rate of modification, and predictability of discharge [137]. In the Anthropocene, all of these distinctive characteristics of lotic systems have induced ecological processes that are influenced by diverse human activities, acting individually, cumulatively, and/or synergistically. Water use may result in considerable ecological impacts [138,139,140,141,142]. Even those permitted by authorized licenses often induce an over 90% decrease in flow discharge, for this motivation strongly influences the main stream habitats (e.g., habitat area, flow velocity, temperature of water, sediments movement, nutrients and primary production, etc.) [143]. Inland water ecosystems, including lotic systems, are in a much more deprived state than, for example, coastal, grassland, or forest ecosystems [118].

There are many significant negative effects on the natural services derived from affected lotic systems, such as the following: increased water extraction and water drainage reflected in terms of lotic and riverine habitat loss; loss of ecosystem integrity through alteration in the timing and quantity of flows, water temperature, sediment and nutrient natural dynamics, bank and delta replenishment, blocks in nutrient cycles and aquatic organism migrations; removal of indispensable elements of aquatic environments; deficit or loss of ecological functions, integrity, habitat cohesion, and biodiversity structure; the reshaping of runoff patterns and dynamics; restriction or obstruction of natural recharge; clogging of aquatic habitats with sediments; loss of living resources and reduction of ecosystem functions; outcompetition of native species, diminishing biomass production and biodiversity, changing chemistry of water and sediments; modification of the natural runoff patterns, increasing erosion, degradation water quality by increasing temperature, modifications in water flow volume in space and time, etc. [144].

It is obvious in this general global context that virtually all ecosystem functions (i.e., habitat, production, regulation, etc.) are under different degrees of socioenvironmental risk. The following elements are under threat: water quantity and quality, habitats, floodplain productivity, fisheries, delta economies, natural flood regulation, wildlife habitats, recreation areas, food production, bank stability and security, water dilution and self-cleaning capacity, sediment and nutrient transport, etc. All of these have a major impact on human communities living adjacent to and depending on lotic systems, including the following: increased health risks, decreased quality and quantity of water, reduced food production and security, reduced economic productivity, reduced household safety, reduced recreational, cultural, historical, and religious importance, increased risk of natural and human-induced calamities, reduced organism genetic diversity and resilience, reduced productive aquatories and territories, etc. [144].

Available information about aquatic, semiaquatic, and riverine biodiversity is fragmentary, with decreasing qualitative and quantitative trends, with a few partial descriptive global overviews of particular taxa and some more detailed regional inventories. The conservation status of these organisms has not been comprehensively assessed, except for certain areas and taxa.

In spite of the fact that humans need to have enough healthy and safe water and water-associated products and resources from both qualitative and quantitative perspectives, and that it was established that the products and services provided by the lotic systems are vital for human wellbeing and poverty alleviation [145,146], the co-evolution of lotic systems and human society shows inherent synergy and feedbacks that are not yet understood [147]. This fact shows that we have unreliable and inconsistent information and knowledge gaps at local, regional, continental, and global scales, which still need to be resolved to really comprehend the present state of lotic systems and the main threats and risks related to human impacts. It also points to the necessity for general access to reliable ecological information. This would allow for the development of adequate mitigation and management strategies, which should be state-of-the-art and adapted from one site/situation to another. Without such work, additional deterioration in terms of the ecological integrity of lotic ecosystems is to be anticipated in the immediate future.

#### 6.2.2. Habitat Fragmentation and Loss—Freshwater Wetlands

The word wetland is used for numerous seasonal and permanent inland freshwater types: springs, creeks, rivers, streams, waterfalls, estuaries, deltas, lakes, oxbows, marshes, swamps, pools, ponds, waterlogged areas, sloughs, potholes, seasonally flooded meadows, sedge marshes, nonforested peatlands, bogs, fens, temporary waters from snowmelt, oases, karsts, etc. There are also many inland freshwater human-made wetland types: ponds, tanks, irrigated land, seasonally flooded agricultural land, water storage areas, reservoirs/barrages/dams/impoundments, excavations, pits, pools, wastewater treatment areas, basins, canals, ditches, etc.

Wetland fragmentation, contraction, alteration, and loss represent significant stressors impacting aquatic environment abiotic and biotic elements [148,149,150,151].

Freshwater wetlands obviously subsume a great deal of variability in terms of the environmental and geological contexts. While certain human activities may promote the development of new wetland ecosystems, such as the construction of hydroelectric and flood control river dam systems, many human activities conflict with wetland ecosystems either directly or indirectly [152,153,154]. A global perspective on wetland salinization is the ecological consequences of a growing threat to freshwater wetlands. In many cases, wetland reclamation involves the construction of levees or other barriers to water flow, and either passive or active systems for removing water from the reclaimed area. While the earliest wetland reclamation efforts are thousands of years old [155], this practice has intensified dramatically in the last two centuries.

Many freshwater wetland geological contexts, such as floodplains and deltas, are characterized by high levels of soil fertility, and therefore have been attractive targets for agricultural activities [156]. Additionally, many major urban centers are located adjacent to large freshwater wetland areas. For reasons ranging from flood risk to mosquito-borne disease to development demands, such wetland areas have also been frequent targets of reclamation [157]. Many major human activities have had indirect consequences for freshwater wetlands [158], such as the overuse of water supply for agricultural activities, the depletion of groundwater, the paving of ephemeral stream channels, deforestation, pollution, etc. Finally, human-induced climate change is increasingly responsible for the destruction of freshwater ecosystems through a range of vectors, including evaporation, increasing salt/chemical loads, saltwater intrusion, etc.

Freshwater wetland fragmentation and loss is ecologically problematic in numerous dimensions. To begin with, freshwater wetland ecosystems play crucial roles in terms of their overall biological productivity and in their articulation with other terrestrial, alluvial/lacustrine, and marine ecosystems. The productivity of freshwater wetlands leads them to be hotspots of biological diversity. This means that freshwater wetland ecosystems to play key roles in the reproduction and life cycles of a huge range of mobile animal species, ranging from global-scale bird migration to upstream fish spawning. Next, freshwater wetland ecosystems are usually very delicate and highly sensitive to external disruptions. Freshwater wetland fragmentation often disrupts the mobility and breeding strategies of keystone animal species that migrate between wetland areas. In a more general sense, the loss of wetland habitat puts particular pressure on organisms at higher trophic levels, which can have major systemic consequences both within freshwater wetlands and without.

Finally, for millennia, human societies have been drawn to the biological productivity of freshwater wetlands. In the past, human hunter-gatherer groups integrated freshwater wetlands into complex mobility patterns and settlement systems, and utilized the many periodic food resource abundances as key elements of their economic life ways [159,160]. In the modern world, freshwater wetlands support a diverse range of fishing communities, such as the Marsh Arabs of the lower Tigris and Euphrates deltas in Iraq [161]. In such instances, constructing flood control and wetland reclamation systems can become the site of intense political conflict having to do with economic production, cultural identity, territorial access, etc.

## 7. The Unsustainable Exploitation/Overexploitation of Freshwater Ecosystem Biological Resources

A biological resource is a matter, thing, microorganism, plant, or animal vital for an organisms’ growth, survival, and reproduction. Resources can be used by an organism, and as a consequence become unavailable to other organisms [162].

Biological resources in freshwater ecosystems have historically been an important source of food for humans. On the other hand, freshwater ecosystems are among the most threatened ecosystems on the planet today. Fish, crustaceans, and mollusks are the resources most used by humans from inland waters as a source of proteins.

At the global level, fish are exploited the most, so freshwater fishing, especially in some less-developed areas (Latin America, Africa, and Asia), is still an important activity that provides the population with a significant amount of food [163,164,165]. In economically developed countries of the world (USA, Europe, Japan), recreational fishing is the dominant form of the exploitation of fish resources, but also a significant source of income (fishing tourism, sport fishing, etc.).

With a trend of global increase in human needs, the danger of excessive use of biological resources, primarily fish, is also increasing. The danger of overfishing also comes from increasing competition for freshwater at a global scale. Seawater does not have this problem or it is of minor importance; however, the use of freshwater in industry, agriculture, power generation, and as drinking water directly affects the decline in fish resources and freshwater fisheries [166]. Almost all inland freshwaters in the world face the problem of overfishing, which is often exacerbated by illegal fishing, especially in less economically developed areas [167,168,169,170,171,172,173]. In the U.S. state of Alaska, where there is commercial fishing in coastal areas and recreational fishing in the interior, there is a problem of overfishing of Pacific salmon during migration [174]. In Scandinavian countries (Norway, Sweden, and Finland), overfishing is not singled out as a factor endangering fish resources [171], in contrast to the acidification of Scandinavian lakes [175] and habitat degradation for salmonid fish species [171]. In contrast, in Russia, fish resources are threatened by overfishing of valuable fish species and pronounced illegal fishing [176]. Illegal fishing significantly complicates the assessment of the total fishing pressure on the fish stock in the inland waters of Russia [176].

Overexploitation of fish reserves, in combination with pollution pressure and the introduction of non-native species, particularly sea flounder (*Petromyzon marinus*) in the 19th and 20th centuries, induced a decline in the fish reserve of the great lakes of North America [177]. An important advance in addressing the crisis was made in the U.S., so that the principle of maximum sustainable harvest (MSY) was replaced by the concept of optimal sustainable harvest (OSY). In some Western European countries, there are no important overexploitation-related issues. Management of fish resources based on this ecosystem principle and postulations of the WFD, as well as strict enforcement and compliance with fishing laws, is a strategy that has led to encouraging results in the conservation of fish resources [169,170,171,172,173,174,175,176,177]. The great wealth of fish stocks in the Amazon basin, tropical Africa, and Southeast Asia are facing overfishing due to the constant growth of the population and effects on living standards [178]. Nonetheless, for these regions, overexploitation of fish stocks due to the wealth of fish is still less important compared to habitat damage, contamination, and climate modification. China has perhaps the most important position in the world in terms of the quantity of freshwater fish caught, but it is experiencing a decline in biomass and the abundance of its most significant fish species for the last four decades [167]. The Danube watershed, particularly in its middle and lower course, faces overfishing, mostly at the level of targeted fishable species. First and foremost, from the Acipenseridae family, the stationary fish species *Acipenser ruthenus* is endangered, mainly due to the pressure on juveniles [119,179,180].

## 8. Invasive Species, Parasites, and Disease Issues in Freshwater Systems

Invasive species, including introduced, alien, and exotic species, are any non-native species that drastically change or disrupt the habitats they colonize [181,182]. From an ecological point of view, one of the most important causes of freshwater changes is biological invasion [183]. Aquaculture, sport fishing, the biological control of mosquitoes and water plants, the aquarium and pet trade, education and research activities, accidents, and illegal introduction are the main reasons for invasion [184,185].

The organism groups that constitute this biological invasion are diverse, including, for example, mollusks, plants, crustaceans, and pathogens. Fish are one of the most important organisms living in freshwater ecosystems. Bernery et al. [183] state that there are 551 established non-native freshwater fish species all over the world and that common carp, *Cyprinus carpio* is the most broadly dispersed fish species.

Native fishes are especially affected by the introduction of alien and invasive fish species [184,185,186,187,188], lowering their reproduction success [189,190,191] and/or harming their ecology through trophic interactions [192]. In addition to these effects, environmental impacts have been reported such as eutrophication and increased nutrient loading [193,194] and reductions in invertebrate populations, such as benthic organisms [195] and zooplankton communities [196].

Invasive species, when they arrive in a new ecosystem, can introduce with them parasites and pathogens [197]. Invasive species are predisposed to be less affected by parasites and pathogens than native species. This may be due to the levels of parasites from the invasive species, the absence of hosts required by the parasites, and the host selectivity of the parasites [198].

The Asian tapeworm, *Schyzocotyle acheilognathi* [199] (Cestoda), is native to Asia. The most well-known and dangerous fish parasite was first described as *Bothriocephalus acheilognathi* from *Acheilognathus rhombeus*, in Japan [200]. This species has been one of the most successful invasive parasites after the introduction of grass carp, *Ctenopharyngodon idella*, to control aquatic plants in the former U.S.S.R., Europe, and North America from China between 1950–1970 [200,201]. Due to the use of bait fishes, minnows, guppies, and Gambusia for mosquito control and sport fishing, irregular displacements between continents have contributed to the broad spread of the parasite [200,201]. Pathological disorders and effects caused by the parasite in host fish have been revealed by various studies [199,202,203]. Korting [204] mentions that infection rates for the parasite and fish mortality reached 100% in fish farms in Germany.

The rosette agent, *Sphaerothecum destruens* (mesomycetozoean), is native to U.S. This intracellular parasite was first discovered in Chinook salmon *Oncorhynchus tshawytscha* [205]. An invasive species of topmouth gudgeon (*Pseudorasbora parva*) is a possible carrier in the wide distribution of this parasite [206]. This parasite has caused significant mortality in the endangered European sunbleak *Leucaspius delineatus* [206].

*Anguillicola crassus* is the swim bladder parasite of Japanese eels. This non-native parasitic nematode was introduced to European eels from Asia. Molnár [207] reported that the parasite has caused mass deaths in European eels (approx. 250 tons) with various environmental factors in Van Lake. Molnár [208] determined various pathological damages to the swim bladder wall of European eels infected with the parasite, such as epithelial hyperplasia and hyperemia at an acute stage and inflammation edema and hyperplasia at a chronic stage.

*Centrocestus formosanus* is a digenetic trematode that is native to Asia. It lives in intermediate hosts such as aquatic gastropods, fish, and amphibians, and in definitive hosts such as aquatic birds, mammals, and humans. The parasite affects gill health and causes respiratory problems for host fish [209]. Francis-Floyd et al. [210] estimate the tropical fish losses infested with this parasite at USD 3 million per year (cited by Mitchell et al. [211]). There are also several reports of human infections through zoonosis, with medical importance [212,213].

*Piscirickettsia salmonis* is a bacterial pathogen of salmonids that was first identified in Chile. *P. salmonis* has been reported from Europe to Oceania to South America. This virulent bacterium has been reported in coho salmon with mortality rates of 30–90% in Chile [214] (Bravo and Campos, 1989). The economic value of the detected fish losses was estimated at USD 10 million [215].

The most important factors in the emergence and spread of invasive species are human-induced. With some exceptions, control of invasive species is complex and difficult. For the most part, their effects are still not fully understood and are often underestimated. Biological monitoring programs must be established to prevent and limit the spread of invasive species at national and international levels. The biology and ecology of invasive and native species in the existing ecosystem and their relationships with each other and with their environment must be examined, and a database should be created for effective management and control. Scientific research on invasive species must be intensified. Similar experts should cooperate (zoologist, ecologist, etc.) in the management of invasive species. Those who produce and trade the adults, larvae, or eggs of both aquarium and edible fish should be informed. Effective legislation, including prohibitions and restrictions, should be implemented for fish imported and transferred between countries. Recreational and fishing boats, gear, and equipment used in fishing should not be moved between ecosystems (lakes, rivers, etc). Precautions should be taken in the live bait trade. Existing ecosystems should not be interfered with for mosquito and aquatic plant control, or, if so, the opinion of experts should be consulted in this regard. Public awareness should not be raised by nongovernmental organizations and official authorities.

## 9. Freshwater Ecosystem Vegetation Issues

Wetland vegetation is characterized by a very close and essential relationship with water. The plants that are most linked to the presence of water are hydrophytes: completely floating, floating rooting, or immersed. The amphibious vegetation, on the other hand, is called helophyte, and it can tolerate both periods of immersion and periods of water desperation. Marsh and swamp vegetation is mainly influenced by groundwater and water supply, and can also be influenced by precipitation. Lowland vegetation (planitial vegetation) is that which develops in alluvial plains (major riverbed), even in cases of exceptional floods. This area is generally the most populated area of the globe, hosting the largest areas covered by civil and industrial settlements, crops, and pastures. Spring vegetation is closely linked to the chemism of the water, and consequently to the leaching of the substrate/soil. For these reasons, among the spring vegetation, we can more easily find oligotrophic species and communities (especially in cold springs). These, together with dystrophic and acidophilic formations, are easily threatened by human impacts, such as grazing and the increase in water temperatures, since they can induce changes in the chemism of the waters or in their hydrogeological cycles. In this context, we can also include waters deriving from the melting of glaciers and snowfields. In the vegetation of the springs, we can also find extremophile organisms and communities, linked to high temperatures or to the strong presence of salts (i.e., sodium chloride, sulfurs, etc.), heavy metals, and strong bases or acids. We can also find very eutrophic communities and even communities influenced by the natural presence of particular minerals such as cyanide and arsenic, or even radioactivity or hydrocarbons. These cases are sometimes defined as cases of natural pollution, but in reality, they are only natural characteristics if they are not due to alteration as a consequence of human impact. These environments should therefore be considered as they are in terms of their possible conservation. The definition of pollution should be used only in cases of ascertained alteration of a previous natural state by humans, and only in cases of pollution should we consider remediation, while trying to achieve this with the least possible environmental impact. In such situations, particular species or communities with complex characteristics often become pervasive, and it may be more worthwhile to leave these intact rather than trying to remedy the pollution [216].

In areas where the human footprint is more historical, such as where pastoralism, agriculture, or the exploitation of forests have been practiced, springs have been harmed by human activities. Today, alongside the restoration of fountains (which remain historical, architectural, and cultural landmarks), we should also think more about the ecological function that these fountains can perform. However, we should also think about the restoration or environmental improvement of wetlands and the related vegetation communities (i.e., springs, aquatic, marshy, swamps, riparian) and related ecosystems linked to specific points of water.

There are also environments of cascades and dripping rocks that are important environments, even if they are sometimes forgotten. In these environments, mineral accretions and concretions of organogenic rocks are often formed. Mosses, lichens, and ferns, as well as occasionally carnivorous plants, may occupy such contexts. In these areas, it is even possible to find formations similar to peat bogs or grasslands in vertical arrangements. Grotto environments, while being able to host fewer photosynthetic organisms, can give space to very specialized and often rare fungi and animals.

In these latter environments, and in the environments of springs, in areas subject to livestock grazing or in areas with overgrazing of natural species, it would be useful to provide for their meticulous and constant monitoring, reserving some environments and some significant strips (for example with adequate fences) to avoid the degradation deriving from excessive trampling, artificial eutrophication and other pollution, grazing, and overturning of the ground. This is also the case for the cutting of the forest, for cultivation and other human activities, and for monitoring and protection.

The hyporheic zone has several advantages, providing a habitat and refuge for various species of fish, aquatic plants, and interstitial organisms. These are responsible for educing the concentration of pollutants in surface waters, controlling the flow and exchange between streams and groundwater, and mitigating the water temperature of the waterbody [216]. The hyporheic zone significantly contributes to the removal of pollutants from surface water through the combination of different processes, such as biodegradation, the action of the microbial community (biofilm), absorption, and desorption [217]. The hyporheic flow carries river water and contaminants dissolved in it into the riverbed (downwelling), where they are temporarily retained and transformed through chemical reactions with a consequent reduction in the concentration of chemical agents in the upwelling. Vegetation and dead plant biomass can therefore play a major role in enhancing biodiversity. For this reason, plant formations should be designed and safeguarded.

Riparian vegetation has its own natural place on the banks of water bodies. It can form some vegetation units all along the perimeter of water bodies, with very different physiognomy (i.e., forests, red beds, sedges, rushes, megaphorbietums, grasslands, etc.). These vegetation contexts are the result of a gradient from aquatic to mesophilic environments. In particular, the expression of this vegetation is functionally connected to other elements of fluvial systems and the contiguous vicinity, and it is part of the riparian zone, which is a landscape unit that is open to lotic system fluxes [218].

This type of vegetation is mostly comprised of wetland types, and it is given by a gradient from aquatic up to mesophilic environments. In particular, the expression of this vegetation is linked to variability in the water level of the water body [219,220,221].

There are particular cases where the riparian vegetation is not of the classic wetland type. It is, in this sense, not directly linked to the water level of the reference water body, and consequently it is not of an azonal type, even if this vegetation is physically located on the edge of a water body. This is due to the fact that the vegetation, despite being located on the *ripa* (Latin for bank, shore), grows in particular conditions (e.g., steep slope, strong drainage, very rapid drainage, deep drainage, artificial banks), or it is located above the average maximum water level. In any case, for some reasons, that vegetation could not be sufficiently influenced by the water body. A typical case is given by *Fagus sylvatica* or *Quercus pubescens* formations that cover streams or river portions, even to xeric species that live on the banks of the water bodies, expanding their coverage up to the water table.

In both cases, these forests can be very important for the functioning and environmental quality of water bodies, and therefore they should be considered among the riparian forests. In order to distinguish them and not generate confusion, it is preferable to distinguish two types. For those reasons, riparial (linked to the *ripa* as a place and also as its related water table dynamic) and riparian (linked to *ripa* as location but not sufficiently linked to the water table dynamic) should not be used as synonyms. Instead, they should be used distinctively to underline the first or the second case, as the term riparian could be used in a wider sense. Riparial forests are always riparian, but riparian forests are not always riparial; for example, when they are of zonal species, such as *Fagus sylvatica*, *Castanea sativa*, *Quercus pubescens*, *Quercus ilex*, *Pinus nigra*, *Pinus mugo*, *Picea abies*, *Abies alba*, *Larix decidua*, *Ostrya carpinifolia*, *Fraxinus ornus*, etc. Examples of typical riparial forests are instead things such as *Salix*, *Alnus*, and *Populus* species.

In any case, both of these formations are essential for the balance of water bodies and to ensure hydrogeological stability, each with its different characteristics. Riparian ligneous vegetation formations have a high value in respect to their diversity and abundance of invertebrate and vertebrate species, which is often higher than the surrounding habitats [222,223,224]. Riparian vegetation is naturally subjected to continuous natural multiple stressors (permanently unstable environments) and to very fast dynamics [225,226] and human impacts [227].

Riparian forests are important from a scientific point of view because they have very rapid and fast cycles, and they allow us to observe dynamics that, in other forests, take centuries to complete or manifest. This can therefore be useful for the observation and study of those dynamics that elsewhere (in the zonal formations) can occur much more slowly and with greater difficulty (even after many centuries). These riparian forests can therefore become a sort of model on which to observe and study forests’ ecological and environmental dynamics and on which to think of proposals for active management or passive conservation [228].

There are also peculiar dynamics; for example, when trees belonging to *Salix, Alnus,* and *Populus* species fall, it is not the end of these trees and their formations, but it is the beginning. These trees in fact reach a second stage (more mature, more advanced, more complex) in being able to continue to live after they have fallen. In the other case, the dead wood decomposes easily and becomes a home for animals, insects, fungi, and other plants. This emphasizes the importance of these trees even when dead, standing, or fallen. These peculiarities can also be found in conifer forests even if they regrow less easily when they break without roots.

Riparian vegetation is important not only as a feeding or nesting place but also as a place of refuge, roost, or passage and habitat for animals and mushrooms. Some of these peculiar functions are often forgotten in impact and incidence assessments, as is the case for heronries.

For riparian trees and related wood formations, today the greatest concerns and threats coincide with current policies in favor of biomass [229,230] and economic development, as well as the channeling of waterways, which have been renaturalized in recent decades in response to historical reclamation and canalization efforts [231]. There are those who even propose to channel watercourses that were saved from this practice in the past. Many problems are also derived from the historical and now chronic anthropization (urbanization, land use, land grabbing, and pollution) concerning the banks, alluvial plains, and water catchment areas [232].

Trees are increasingly seen as dangers, and trunks that have fallen into the water are increasingly seen as obstacles to be eliminated in order to reduce risk, to not compromise the water flow, or for exaggerated security concerns. Instead, living trees and fallen trunks (and branches) hold important functions (i.e., as natural bridles, a basis of the food chain, and a precious substrate for biodiversity).

Canalization is therefore a problem linked not only to the rectification of the rivers, but also to the simplification of the banks or their modification in order to favor a greater flow, or to prevent flooding), which can even include cementification, piling, or artificial earth embankments. Canalization must also be seen as an incremental increase in the risk linked to the presence of natural vegetation; therefore, it increases maintenance costs and increases the environmental risk, increasing the need for the removal of fallen branches and trunks, as well as for the removal of vegetation.

On the one hand, this problem is due to the lack of knowledge and preparation of the operators and managers of the bodies in charge, but it is also due to the erroneous policies and subsidies in favor of biodiversity, agriculture, energy, and economic development. This includes the use of virgin woody biomass for energy purposes, and to the land that is cultivated and forests that are harvested mainly to take advantage of funding, facilities, or project funds, often described as being carried out for the sake of the environment or in favor of development. On the other hand, however, we cannot forget the difficulties and problems associated with excessive and incremental urbanization.

For too long, in some sectors of land management and forestry, it was thought that it was good water management to drain the greatest volume of water in the shortest possible time towards the sea; this not only destabilizes water bodies up to the most upstream sections, but it also produces enormous upheavals in the water balance [233,234,235] with implications that are increasingly concerning not only for biodiversity but also for agriculture and people’s lives. Consequently, the retention of the water was conceived only of artificial barrages, which sometimes produced important habitats and other times produced enormous upheavals in the landscape, in the hydrogeological/sedimentological balance and in the water’s characteristics. All of this influenced wetland ecosystems, vegetation, and riparian communities.

To counteract the hydrogeological problem of the slopes, together with the negative consequences of sediment equilibrium (erosion/deposit) in the hydrographic basins, it was necessary to reforest huge surfaces, often with species suitable for surviving where the soil had disappeared. In the points where these reforestations have been able to develop at their best (avoiding cleaning interventions, cuts, thinning, fires), very interesting ecological conditions are created (tall trees, dead wood, developed soil and stands, marked renewal of broadleaved trees, advanced succession, less erosion, greater water retention of the slope basin). Many benefits to the stability of the slopes and the water retention of the hydrographic basins were also given by the decrease in anthropogenic pressure and land use, which allowed greater grassing, and the return of surfaces covered with shrubs and woods. Sometimes, punctual and well-though—out hydraulic works (bridles, jumps, brushes) have been able to make up for the lack of plants, woods, and natural bridles given by fallen branches and trunks, even if only in part. These forestations, when well designed and built, are to be considered important achievements of forestry experts.

Beyond the hydrogeological balance, there are also other threats deriving from the state of the hydrographical basin and by the climate, such as the desertification of reclaimed wetlands due to water extraction, and the diversion of water for production purposes, etc.

Soil impermeabilization (urbanization, concreting-over, etc.) and a lack of vegetation cover has destabilized hydrogeological conditions. Furthermore, the canalization of rivers and the artificialization of the banks have often had the effect of narrowing the banks and therefore increasing the flow rate, and consequently, the erosion in depth, with consequent greater instability of the banks and with greater risks and damage following floods.

Climate change can therefore have a greater impact on these problems while also facilitating the entry and impact of pathogens and drought phenomena. Even the most open plant communities (which can be primary or secondary riparian vegetation) can be very important and can vary a great deal physiognomically (i.e., reed beds, sedges, rushes, grasslands, prairies, pastures, megaforbs, bushes, acid and alkaline peat bogs, swamps, etc.).

These formations are also very sensitive and threatened by channeling and anthropization, as well as by cleaning the vegetation on the banks [236]. All these and the following issues are the same for other types of wetland vegetation such as swamps, marshes, springs, and aquatic vegetation.

We can observe that how we think about the management of wetland plant communities can be in an excessively active way, and therefore we see the paradox of how we intervene on areas with forest potential to maintain or restore secondary reedbeds. Elsewhere, the same formations can have a primary potential when action is taken to conserve or restore pastures or other secondary formations.

Otherwise, correct planning and management should consider managing formations looking on where they can be primary, and therefore requiring fewer costs and less human disturbance for their conservation, and where they can therefore guarantee a greater stability and a greater durability.

With regard to plants, there is also the problem of alien species, which, however, from a plant point of view, should be seen in a less ideological way. It is not unusual to see greater impacts related to the control of these alien species than those caused by the invasive species themselves. We should concentrate more on the study of these species and how communities adapt or restructure themselves (resistance and resilience), focusing more on dynamics and science than on summary judgments, on proposals for eradication, or on intolerance towards non-native plant species.

## 10. War and Water

According to UNO assessments, at the middle of the current century, about 7 billion people in 48 countries will face a water deficit. Of course, climate change increases these risks and the risks of the wars over water resources [237].

Since the first armed conflict over water in about 2500 BC in Mesopotamia [238], water has been a cause of—or weapon in—human warfare [239]. Today, most observers consider efforts to gain access to the Dnieper waterways to provide the occupied territories of Donbas and Crimea as one of the significant reasons of the Russian aggression in Ukraine [240].

Any armed conflict afflicting freshwater resources carries significant threats and risks. First of all, these are threats to population, from the settlements’ flooding owing to the destruction of dams and weirs to the inaccessibility of the drinking water of appropriate quality, etc. For example, in the Vietnam War, American troops bombed dams, and as a result, more than 2 million people drowned or died of starvation. In Kosovo, the Serbs contaminated springs, and in Zambia, the war destroyed the pipeline supplying a city with almost 3 million inhabitants [241].

Aquatic ecosystems are also affected by hostilities. Here are some recent examples from the Russian–Ukrainian war, which has already had and is expected to have an impact on more than 60,000 streams and 20,000 lakes, as well as about 50,500 ponds. The fact of the explosion at a dam, aimed at flooding of the Irpin River flood land on 26.02.22, is well known. In fact, this measure alone provided efficient protection of Kyiv from aggression. Owing to the military engineering activities aimed at the use of the Irpin flood land as a natural barrier for the occupation by Russian troops, and owing to damage of the coastal and riverbed sections and destruction of the hydrotechnical facilities over the hostilities, unfavorable—and even threatening—ecological situations have arisen [242].

The disastrous changes to the river hydromorphology and the deterioration of the water quality are because of the flooding of cesspools and landfills, the direct death of the aquatic living resources, the reduction in aquatic biodiversity, and the degradation of the floodplain landscapes, which remained impounded for a long period. This, in turn, caused fish deaths in the summer. On March 4th, about 7 million m^3^ of water was discharged from the Zhytomyr city reservoir in order to disrupt the pontoon crossing of the Teteriv River and flood the enemy’s equipment. The operation was successful and about ten units were flooded. However, this resulted in oil pollution of the river, and in the drained reservoir, all bottom fauna was killed. The explosion of the Oskil reservoir dam, the buffer water source of the Siverskiy Donets, Donbas channel on 2 April 2022, resulted in the flooding of some settlements and the denudation of its bed. Besides the direct loss of the aquatic resources and biotopes for the phytophilous fishes and invertebrates, as well as the transition of the limnophilous communities to the regime of high water flow, one more problem has arisen: about 120 km^2^ of the bed sediment has been denudated, which is easily transported by the wind [243].

The contamination of water bodies is a particular problem. Besides the direct destructions of the wastewater treatment plants, any settlement close to the hostilities suffers from disorders of the electric current supply, which results in turning off the pumping facilities and stopping the air supply. Therefore, the polluted waste waters are discharged directly into the rivers untreated. Even more catastrophic problems come from the halting of the pumping of mine waters. Instead of entering the tailing pond, they directly flow into the rivers. In fact, practically the entire basin of the Siverskyi Donets River and rivers of the northern coast of the Sea of Azov are affected by the mine waters. The environmental effects can be assessed only after the hostilities’ cessation by comparison of data with those obtained before the wide-scale Russian aggression [244].

Therefore, monitoring of the water bodies is the most essential element of their management, even during the war. One more important issue consists in assessment of the ecosystem services of the hydroecosystems. The preliminary assessment, carried out according to the GEF methodology of the ecosystem services evaluation [245] has shown that losses from the disordered level regime of the Kyiv reservoir, caused by the weir explosion in the Irpin River mouth, and further draining of the shallow areas, as well as the degradation of spawning areas and the death of the early-spawning fish eggs, have amounted to about EUR 23 million [243].

On the whole, the algorithm of actions in water management over the war is reduced to determine the actual ecological state of the modified, destroyed, and contaminated water bodies. The correction of river basin management plans and insertion of renaturalization and rehabilitation measures, as well as the assessment of losses from unobtained ecosystem services from the aquatic ecosystems, are the basis for the appeal in the UN International Court of Justice.

Taking into account global climate change and probable armed conflict in the future, it is vital to carry out an inventory of the potential impassable river sections and overwatered flood lands in the development of a methodology for their rehabilitation and conservation. This is true with respect to this issue of biodiversity on the one hand and as a natural barrier to restrain the troop advances on the other. Along with the reconstruction of wastewater treatment plants, it is possible to develop and implement technologies of natural riverbed rehabilitation, with the potential for the free meandering of stream systems in floodplains, phyto- and ichthyoamelioration, the building of new hydrotechnical facilities with biopositive properties, and progressive fish passes over the reconstruction of the destroyed dams, in the overflows of the exploded bridges, etc.

## 11. Conclusions

This paper is a synthetic overview on some of the threats, risks, and integrated water management elements in freshwater ecosystems and its provisions for human needs and water conservation elements related to freshwater: (1) introduction and background; (2) water basics and natural cycles; (3) freshwater roles in human cultures and civilizations; (4) water as a biosphere cornerstone; (5) climate as a hydrospheric ‘game changer’ from the perspective of freshwater; (6) human-induced stressors’ effects on freshwater ecosystem changes (pollution, habitat fragmentation, etc.); (7) freshwater ecosystems’ biological resources in the context of unsustainable exploitation/overexploitation; (8) invasive species, parasites, and diseases in freshwater systems; (9) freshwater ecosystems’ vegetation; (10) the relationship between human warfare and water. All of these issues and more create an extremely complex matrix of stressors that plays a driving role in changing freshwater ecosystems and their capacity to offer sustainable products and services to the human society. Internationally integrated policies, strategies, assessment, monitoring, management, protection, and conservation initiatives alone can diminish and hopefully prevent the long-term deterioration of Earth’s freshwater resources and associated resources.

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
