# Peer review of "Freshwater as a Sustainable Resource and Generator of Secondary Resources in the 21st Century: Stressors, Threats, Risks, Management and Protection Strategies, and Conservation Approaches"

_ijerph, 2022, doi:10.3390/ijerph192416570_

Round 1

Reviewer 1 Report

The overview is interesting scientifically but has dozens of grammar and spelling mistakes, which affected negatively how to grasp the real meaning of the author. Some low level errors include mistaking Water Framework Directive (WFD) as FWD, offer as ofer, paintfull as paintfull, hopefully as hopefullz, etc.  in the manuscript.

Human induced causes on natural water deterioration might include urbanization in addition to population growth, economic development etc. It should not be neglected and needs to be discussed. 

The category of POPs seemed confusing. According to the Stockholm  Convention, the POPs family could be divided by intentional and Unintentionally Produced POPs, or legacy POPs and Newly-listed POPs (Emerging POPs), depending on different way of taxonomy.

The references and citations are also not standardized and consistent. 

Author Response

1. The overview is interesting scientifically but has dozens of grammar and spelling mistakes, which affected negatively how to grasp the real meaning of the author. Some low level errors include mistaking Water Framework Directive (WFD) as FWD, offer as ofer, paintfull as paintfull, hopefully as hopefully, etc.  in the manuscript.

THE GRAMMAR, SPELLING AND WRITING ISSUES WERE CHECKED AND CORRECTED BY THE ENGLISH NATIVE LANGUAGE AMERICAN RESEARCHER/CO-AUTHOR OF THE PAPER.

2. Human induced causes on natural water deterioration might include urbanization in addition to population growth, economic development etc. It should not be neglected and needs to be discussed.

PLEASE SEE BELOW THE ADDED TEXT AND COMMENTS.

URBANIZATION ISSUE WAS DISCUSSED IN THE POPULATION GROWTH AND ECONOMIC DEVELOPMENT; PLEASE SEE BELOW. URBANIZATION WAS NOT APPROACHED IN EXTENSO IN THE PAPER AS A DISTINCT SUBCHAPTER BECAUSE IT IS TOUCHED IN THE PRESENTED SUBCHAPTERS LIKE FOR EXAMPLE: Human induced stressors drivers’ effects on freshwater ecosystems change (pollution, habitat fragmentation, freshwater ecosystems biological resources unsustainable exploitation/overexploitation; etc. In the authors opinion urbanization is a complex issue which need a separate paper approach regarding its relation with freshwater, with a different designed paper structure and data.

NEW IMPROVED TEXT ADDED

In this circumstances guaranteeing a sufficient amount of clean water for peoples’ needs become a precondition for appearance and accelerated and extended flourishing urbanization, state formation, and all the progress in the human society organization which came after, all of these determining huge changes in demographics. [32, 33].

3. The category of POPs seemed confusing. According to the Stockholm  Convention, the POPs family could be divided by intentional and Unintentionally Produced POPs, or legacy POPs and Newly-listed POPs (Emerging POPs), depending on different way of taxonomy.

THIS INFORMATION WAS COMPLETED AS IT WAS SUGESTED; PLEASE SEE BELOW

Persistent organic pollutants (POPs) are organic chemical compounds with toxic characteristics, resistant to decomposition, which bioaccumulate in living organisms and can be moved and carried by air, water and migratory organisms at very long distances [94]. In general, according to the Stockholm Convention, the POPs family could be divided by intentional and Unintentionally Produced POPs, or legacy POPs and Newly-listed POPs (Emerging POPs).

POPs can have two categories of origin – dibenzofurans and dioxins, chemical compounds released by volcanic manifestations and fires [95, 96] or resulted from the humans activities – different chemical substances used in agriculture like pesticides (hexachlorobenzene, chlordane, heptachlor, DDT, aldrin, dieldrin, endrin, mirex, toxaphene, etc.), a high variety of industrial chemical substances (perfluorinated compounds, brominated compounds, polychlorobiphenyls, hexachlorobenzene, etc.) or not deliberately or on purpose created, by-products of industrial activities or eliberated in environment by some materials burning (dioxins and furans) 97-100.

4. The references and citations are also not standardized and consistent. 

THE REFERENCES AND CITATIONS WERE STANDARDIZED AND IMPROOVED AS SUGESTED.

Reviewer 2 Report

This manuscript describes some threats, risks, integrated water management elements in freshwater ecosystems and their provisions for human needs, as well as freshwater related water-saving elements. This manuscript also highlights the importance of water conservation and rational distribution of water resources. However, there are some mistakes in sorting, grammar and spelling. Next are my specific comments:

1)The introduction of this article does not mention the importance and novelty of writing this article. This is important.

2)Some references cited are outdated, such as the 85th and 89th references.

3)In line 25, “efects” should be “effects”.

4) In line 30, “ofer” should be “offer”.

5) In line 41, understanding” should be “understand”.

6) In line 64, highlight” should be “highlighting”.

7) In line 70, ecologycal” should be “ecological.

8) In line 83, staus” should be “status”.

9) In line 102, “ilnesses” should be “illnesses”.

10) In line 129, “globall” should be “global”.

11) In line 141, “belive” should be “believe”.

12) In line 169, “bodyes” should be “bodies”.

13) In line 179, “interelation” should be “interrelation”.

14) In line 183, “deply” should be “deeply”.

15) In line 188, “usuall” should be “usual”.

16) In line 189, “Sush” should be “Such”.

17) In line 199, “allong” should be “along”.

18) In line 219, “exrtaordinary” should be “extraordinary”.

19) In line 226, “comerce” should be “commerce”.

20) In line 238, “biger” should be “bigger”.

21) In line 256, “photosynthesys” should be “photosynthesis”.

22) In line 412, “powerfull” should be “powerful”.

23) In line 416, “suplementary” should be “supplementary” and “forcasted” should be “forecasted”.

24) In line 496, “molusk” should be “mollusk”.

25) In line 536, “paterns” should be “patterns”.

26) In line 565, “necesity” should be “necessity”.

27) In line 702, “rhombeus” should be “rhombus”.

28) In line 718, “nematod” should be “nematode”.

29) In line 903, “comcreeting” should be “concreting”.

30) In line 984, the title number should be nine.

31) In line 1059, the title number should be ten.

32) In line 1068, “ilnesses” should be “illnesses”.

33) In line 1074, “hopefullz” should be “hopefully”.

Author Response

There are some mistakes in sorting, grammar and spelling. Next are my specific comments.

THE SORTING, GRAMMAR AND SPELLING ISSUES WERE ADRESSED.

1)The introduction of this article does not mention the importance of writing this article. This is important. PLEASE READ THE ADDED TEXT BELOW.

In spite of the fact that the water resource global management is extremely complex, owing to diverse geophysical, climatic, socio-economic, and political realities, the core idea of this approach should be the act that only the conserving the complex ecosystems structures and functions can solve the present Global water crisis! In this context, the importance of this review article, is to bring together a series of main common stressors with synergic effects, to be considered in freshwater complex ecosystems assessment, monitoring, and management.

Round 2

Reviewer 1 Report

The refined text seems ok now.

Reviewer 2 Report

This manuscript can be accepted.